# COVID-19 Vaccination Status among Pregnant and Postpartum Women—A Cross-Sectional Study on More Than 1000 Individuals

**DOI:** 10.3390/vaccines10081179

**Published:** 2022-07-25

**Authors:** Urszula Nowacka, Paulina Malarkiewicz, Janusz Sierdzinski, Aleksandra Januszaniec, Szymon Kozłowski, Tadeusz Issat

**Affiliations:** 1Department of Obstetrics and Gynecology, Institute of Mother and Child, Kasprzaka 17a, 01-211 Warsaw, Poland; aleksandra.januszaniec@imid.med.pl (A.J.); szymon.kozlowski@imid.med.pl (S.K.); tadeusz.issat@imid.med.pl (T.I.); 2Department of Obstetrics and Gynecology, School of Medicine, Collegium Medicum of the University of Warmia and Mazury, Al. Warszawska 30, 10-082 Olsztyn, Poland; p.malarkiewicz@gmail.com; 3Department of Medical Informatics and Telemedicine, Medical University of Warsaw, Litewska 14/16, 00-581 Warsaw, Poland; jsierdzinski@wum.edu.pl

**Keywords:** COVID-19, pregnancy, vaccination, vaccine uptake, vaccine hesitancy, Tdap vaccine, vaccine, vaccination status, fetus, maternal–fetal medicine

## Abstract

Pregnancy is a well-known factor for vaccine hesitancy and immunization remains the most effective form of prevention against coronavirus disease (COVID-19) related complications. The objective was to estimate vaccine uptake and hesitancy rate, characteristics, and factors contributing to a decision-making process among pregnant and postpartum individuals. This was a prospective cross-sectional study on 1033 pregnant (54.1%) and postpartum (45.9%) women conducted between December 2021 and March 2022 in a tertiary center for maternal–fetal medicine. Logistic regression was used to assess characteristics related to the vaccination decision process. Among responders, 74% were vaccinated and 26% were hesitant (9% planning to vaccinate and 17% totally opposed). Only 59.8% were offered a vaccine by healthcare professionals. Women with higher levels of education (OR 2.26, *p* < 0.0001), who received positive feedback about vaccination (OR 2.74, *p* = 0.0172), or were informed about COVID-19 complications in pregnancy (OR 2.6, *p* < 0.0001) were most likely to accept the vaccination. Hesitancy was associated with multiparity (≥3, OR 4.76, *p* = 0.006), worse educational status (OR 2.29, *p* < 0.0001), and lack of previous COVID-19 infection (OR 1.89, *p* < 0.0001). The most common reason for rejection was insufficient safety data (57%). Understanding factors behind vaccination status is crucial in lowering complications in mothers and newborns and targeted action may facilitate the uptake.

## 1. Introduction

Since the beginning of the coronavirus disease (COVID-19) pandemic, pregnant individuals have been assigned to a high-risk group due to resemblance to previous respiratory infection outbreaks, such as severe acute respiratory syndrome (SARS), Middle East respiratory syndrome (MERS) 1–3, and influenza [1]. Various viral infections, which pregnant women are susceptible to, may contribute to unfavorable pregnancy outcomes through numerous molecular mechanisms [2,3]. Severe acute respiratory syndrome coronavirus 2 (SARS-CoV-2) infection in pregnant women has been linked not only to an increased risk of obstetric complications such as preterm birth and stillbirth but also to hospital and intensive care unit admissions, need for mechanical ventilation, and death, compared to non-pregnant age-matched group [4,5]. This risk can be further multiplied by the presence of chronic comorbidities [5]. Immunization against SARS-CoV-19 remains the most effective form of prevention against COVID-19-related complications [6].

There are two major benefits of vaccinating women in pregnancy: it protects a woman from diseases that she can be particularly susceptible to in the course of gestation, and, indirectly, protects the developing fetus [7]. Additionally, it triggers antibody production, which is transferred through the placenta (IgG) or secreted with breastmilk (IgA) to protect the fetus and infant within the first months of life [7]. The majority of vaccines are not administered until an infant reaches at least 6 weeks of life; therefore, maternal immunization may fill the gap when a baby is immunocompromised [7].

Although initially pregnant women were not offered vaccines due to safety concerns, guidelines from the Centers for Disease Control and Prevention were shortly thereafter changed in favor of immunization [8,9]. The UK recommendation was modified on 30th December 2020 advising vaccine uptake for pregnant women at greater risk of contracting an infection (healthcare workers, frontline staff) or with concomitant risk factors [10]. Shortly thereafter, pregnant individuals were included in vaccination schedules with the rest of the population, and the mRNA vaccines were the preferred products to be administered [10]. The Polish Society of Obstetricians and Gynecologists issued consistent local recommendations on 21 April 2021 [11]. Currently, the majority of the guidelines worldwide indicate that pregnant individuals should be offered the vaccination at the same time as non-pregnant women based on their age and clinical risk group [12]. Having COVID-19 vaccination in pregnancy does not alter perinatal outcomes [13].

COVID-19 vaccine effectiveness in pregnant women is comparable to non-pregnant individuals and the safety data was derived from animal studies as well as from women unaware of pregnancy at the time of vaccination [12]. Moreover, there are emerging pharmacovigilance reports showing no additional adverse events or complications after receiving the vaccine in gestation, with more than 200,000 pregnant individuals vaccinated in the USA to date [14]. Vaccine coverage is known to be substantially lower in pregnant women than in the general female population [12]. Gestation is a well-known factor for vaccine hesitancy, defined as uncertainty or refusal of a vaccine, despite the availability of vaccine services [5] Although professional organizations underline the need for appropriate counseling and shared decision-making provided by healthcare professionals, only about half of the pregnant individuals in the USA are vaccinated for influenza and pertussis every year [5]. Factors associated with hesitancy are minority ethnicity and race, low socioeconomic status, and young age [5,8]. Understanding the lack of acceptability of the COVID-19 vaccine and addressing certain issues is crucial in lowering morbidity and mortality rates in pregnant women, as 98% of expecting individuals admitted to the hospital because of COVID-19 in one of the studies were unvaccinated [15].

The majority of databases do not indicate the number of women vaccinated during pregnancy and very few studies investigate vaccine hesitancy within pregnant individuals [16]. Therefore, the objective of this multi-method study was to investigate vaccine uptake among pregnant and postpartum women as well as associated characteristics and decision-making concerns.

## 2. Materials and Methods

This was a prospective cross-sectional study about COVID-19 vaccination in pregnancy conducted in a tertiary center for maternal–fetal medicine and obstetrics. Recipients of all sorts of care (emergency services, online consultations, midwifery care, general obstetrics, and maternal–fetal medicine) provided by the site were offered to take part in an anonymous survey. Depending on patients’ preferences and a form of consultation, a paper form or an online link was provided. The survey assessed sociodemographic factors and perceptions about vaccination for COVID-19, influenza, and pertussis (tetanus, diphtheria, and acellular component of pertussis—Tdap), as recommended by local guidelines. The questionnaire addressed uptake, views on vaccination in pregnancy, prior exposure, and critical factors in the decision-making process. The survey was open to eligible beneficiaries of care provided by the Institute of Mother and Child in Warsaw, Poland, between December 2021 and February 2022. The study was approved by the Institute of Mother and Child Research Ethics Committee (Appendix to Approval No 49/2020).

### 2.1. Participants

All the participants were ≥18 years old. The inclusion criteria were women fluent in their native language with a singleton, uncomplicated pregnancy, and 10–42 weeks’ gestation confirmed by a healthcare professional, or up to 6 weeks postpartum. All the women were invited to fill in an anonymous questionnaire. Participants gave written informed consent (paper version) or agreed to undergo an online survey, which had a consent part with a tick box, to take part in the study.

### 2.2. Recruitment

Eligible participants were recruited in person or while telephone calls by the research staff. Before claiming a voluntary wish to participate, every woman was described with a purpose and formula. A piece of information revealing a thorough description of the study, the main researcher’s bio, and contact details in case of any further queries, feedback, and complaints was placed on the cover page of the folder, followed by written consent (paper version). The online questionnaire was prefaced with an informational part about the study purpose and data management and a tick box to consent for the study. Therefore, all of the enrolled participants gave informed consent. All the partially filled forms (lacking date of birth and pregnancy dating status) were discarded from further analysis. No extra appointments were scheduled after the completion of the questionnaire.

### 2.3. Design

The survey formula was designed by a multidisciplinary team that mainly consisted of midwives and physicians and was based on our previous study on generalized anxiety in the pandemic [17]. The survey had two parts:

Demographic part: questions related to demography, pregnancy order, education, socioeconomic factors, profession, chronic conditions, cigarette use;

Vaccination part: questions related to uptake, personal views on COVID-19 vaccination, healthcare professionals’ and women’s environment’s influence, influenza, and pertussis vaccination status.

All the data was collected anonymously, paper questionnaires were stored separately from the consent forms. There were 24 questions in total, the number of questions answered depended on the vaccination status. Vaccine hesitancy was previously defined by the World Health Organization as uncertainty or refusal of vaccination, despite the availability of vaccination services [5]. Previous studies focused mostly on hesitancy using a Likert scale [5,8]. To keep the survey as simple and short as possible and also to avoid abandoning it while filling in, we decided on omitting the mentioned part and focused on motives in favor of and against the vaccination. Because our center reports less than 5% of the patients of non-Caucasian origin in general, the race and ethnicity issue was not raised.

### 2.4. Outcomes

The main objective of the study was to quantitatively assess the vaccination status in pregnant and postpartum women. Secondary outcomes were factors contributing to the decision-making process, such as the influence of healthcare providers and the professional bodies, reasoning for not choosing or choosing the vaccination, as well the time of receiving the vaccination doses (pre-, during, and post-pregnancy). Individuals could choose more than one reason. The survey also focused on sociodemographic factors and views on influenza and pertussis vaccination. The following patient characteristics were assessed: age, parity, place of residence, education, current employment status, financial situation, number of people in the household, chronic diseases, and tobacco use. The second part addressed the “3 Cs” model (complacency, convenience, and confidence) determined by the WHO Vaccine Communications Working Group in order to understand the determinants of vaccine hesitancy [18]. The assessment included vaccine acceptance, time of administration of the first dose (pre-, during, and post-pregnancy), the total number of doses, COVID-19 exposure, reception of the vaccine by friends and family, and opinion of the healthcare providers (1–4 Likert scale), counseling about the benefits of vaccination, influenza, and pertussis vaccine uptake. The questions regarding factors contributing to the decision-making process were multiple-choice with a possible self-proposed answer.

### 2.5. Statistical Analysis

All sociodemographic and clinical factors were compared between women who accepted the vaccine and those who rejected it [19]. Logistic regression was used to calculate odds ratios (ORs) between patient characteristics (parity, place of residence, education, history of COVID infection, proposal of vaccination by healthcare professionals, attitude of healthcare professionals towards vaccination, information about risk of infection in pregnancy from healthcare professionals) and COVID-19 acceptance variables [20]. We did not fill in the missing data. Data analysis was performed using SAS statistical software version STAT 9.4 (SAS Institute, Cary, NC, USA). *p*-values < 0.05 were considered significant and all tests were two-tailed.

## 3. Results

During the study period, 1033 participants finished the survey—350 papers (33.9%) and 794 electronic (66.1%) forms were completed. A total of 559 (54.1%) women were pregnant and 447 (45.9%) were postpartum (not discharged from hospital after delivery) while completing the form. The group was divided into the following subgroups (Figure 1): (1). vaccinated before pregnancy; (2). vaccinated while pregnant; (3). vaccinated before pregnancy and booster administered while pregnant; (4). planning to vaccinate; (5). unvaccinated. As a result of thorough consideration, we decided to not include in logistic regression individuals planning to vaccinate as their status and the final decision could not be established. Therefore, for further logistic analysis, the following groups emerged: vaccinated (1 + 2 + 3) and unvaccinated (5). The demographics behind the study group are presented in Table 1 and Figure 1.

The mean age of participants was 30.5 years (SD 4.1). A total of 40.5% were living in very large urban centers with more than 500,000 inhabitants. A total of 68.2% had a master’s degree and almost all (97.5%) reported their financial status to be very good or sufficient to cover all their expenses. Almost three-quarters (72.2%) constituted office and administrative workers and nearly half (45.4%) were on sick leave while pregnant. A total of 34.9% reported some kind of a chronic illness with hypothyroidism being the most common (17.2%). A total of 1.3% reported tobacco usage while pregnant. A total of 37.7% of responders had a history of COVID-19 infection with a quarter (26%) reported during pregnancy. A total of 74% of responders were vaccinated against COVID-19 (16% before pregnancy; 40% during pregnancy; 18% before pregnancy with a booster while pregnant); 26% were hesitant (9% planning to vaccinate and 17% totally opposed). For women vaccinated in pregnancy, the second trimester was the most common time for administration (43.6%). Only 59.8% of individuals were offered a COVID-19 vaccination by their doctors or midwives but only less than 1% (0.8%) were actively discouraged from vaccination by a healthcare professional. A total of 54.6% claimed that the decision about vaccine intake was totally a personal one, 43.7% experienced negative feedback about vaccination in pregnancy from relatives or friends, and 5.6% from healthcare professionals. A total of 56% of unvaccinated responders rejected vaccination due to insufficient clinical data and only 12.1% due to a potential influence on the developing fetus. The majority of individuals (60.1%) were not offered a Tdap/influenza vaccine and only 13.9% of postpartum women completed both Tdap and influenza vaccination. Less than half (48.9%) were informed about the increased risk of COVID-19/pertussis/influenza infection in pregnancy.

With regards to logistic regression, factors associated with higher odds of vaccine acceptance were nulliparity (OR 1.74; 95% CI 1.43–2.12), living in a large or a very large city (OR 1.51; 95% CI 1.09–2.1), higher education (master’s and bachelor’s, OR 2.26; 95% CI 1.53–3.34), history of COVID-19 infection (OR 1.92; 95% CI 1.43–2.59), a vaccination offer from a doctor or midwife (OR 1.54; 95% CI 1.11–2.12), positive attitude of a doctor/midwife towards the vaccination (OR 2.74; 95% CI 1.2–6.29), and information received from a doctor/midwife about a high risk of COVID-19 complications (OR 2.146; 95% CI 1.55–2.8) (Table 2). Factors associated with higher odds of rejection were having children: 3 versus 0 (OR 4.8; 95% CI 2.0–11.6), 3 versus 1 (OR 2.15; 95% CI 0.89–5.2), 3 versus 2 (OR 1.78; 95% CI 0.69–4.59), lower educational status (elementary + vocational + high school) (OR 2.29; 95% CI 1.55–3.39), and lack of COVID infection in the past (OR 1.89; 95% CI 1.4–2.55) (Table 3)

## 4. Discussion

### 4.1. Main Findings

There are a few main findings of this study: (1) the majority of pregnant/postpartum women in the sample were fully vaccinated, (2) in 40% of cases, the COVID-19 vaccine was not offered by a healthcare professional, and only half were informed about the increased risk of severe infection, (3) recommendations issued by the professional body helped to convince almost a quarter of women, (4) the most common reason for vaccination rejection was insufficient data, (5) in 60% of cases, the influenza/Tdap vaccine were not offered by healthcare professionals, (6) factors associated with vaccine acceptance were nulliparity, living in big urban centers, higher education, history of COVID infection, vaccination offer from a doctor/midwife, positive attitude of a doctor/midwife towards the vaccination, and information received from a doctor/midwife about the high risk of COVID-19 complications, and (7) factors associated with rejection were having children, lower educational status and lack of COVID infection in the past.

Our study of 1033 women investigated attitudes towards the COVID-19 vaccine during the fourth wave of the pandemic, at the time when the Omicron variant dominated, almost a year after immunization in pregnancy have been introduced. Important conclusions from studies had been already available at the start of the survey—almost all hospitalized pregnant individuals were unvaccinated and 17% of the total critically ill patients in English hospitals on extra corporeal membrane oxygenation (ECMO) therapy were pregnant women [8].

Our study found quite a substantial level of hesitancy among pregnant and postpartum women, which is consistent with pre-existing data [5,12]. Expecting individuals are likely to be in opposition to vaccination compared to age-matched non-pregnant peers [21]. Moreover, the results are highlighting the role of healthcare providers in vaccine hesitancy. Unfortunately, 40% and 60% of women in the cohort were not offered immunization against COVID-19 and influenza/pertussis, respectively, by their doctors and midwives. The extent of this phenomenon is worrying as the main focus is usually put on patients’ willingness rather than on the information provided by the physicians.

This factor may trigger placing vaccination matters on the interview checklist for doctors and midwives. Educational actions may also play an important role in modifying those unexpected factors, as vaccine hesitancy among healthcare providers is not as rare as expected [22].

The most common reason for vaccine rejection in our study was insufficient data on vaccine safety in pregnancy. Therefore, counseling based on safety for unborn babies and infants may be more effective than emphasizing a disease threat to the mother, an issue previously described [23]. Presenting supportive data on facilitating a newborn’s humoral response and antibodies transfer through the placenta and breast milk may also make hesitancy less probable [24]. Another reason for hesitancy included concerns regarding safety related to insufficient data and lack of reliable publications. It suggests that this factor is modifiable, as around a fifth of individuals could change the decision as more data will emerge from different cohorts over time. In the meantime, pregnant and breastfeeding individuals could be inclined towards a decision using guidelines, which play a great role in a decision-making process, as professional recommendations helped more than a fifth of responders. Healthcare providers should introduce this important matter while counseling, as access to recommendations may not be universal, especially within deprived groups. Ethnic minority background is a well-known factor not only for COVID-19, Tdap, and influenza vaccine hesitancy but also for dying from COVID-19 [5,8]. We are currently experiencing the biggest migration crisis in Europe since World War II—the number of people fleeing from Ukraine is estimated to be almost 4 million at the moment, with the vast majority choosing Poland as a main destination [25]. Only 35% of the Ukrainian population was vaccinated, with boosters administered to only 1.7% [26]. Therefore, as people arriving in Poland are mostly women of reproductive age, we can anticipate an increase in infection numbers among pregnant individuals in the future. This should trigger informational actions at every level of healthcare, encouraging women to undergo the immunization process. Moreover, uptake should be promoted not only within a group without an initial vaccination course—secondary hesitancy related to individuals vaccinated before pregnancy with two doses needs promotion of a booster dose to be taken in the course of pregnancy.

Vaccine hesitancy/negative vaccination status in pregnancy in our study was lower than reported by other authors (37.9% according to Skirrow et al. [8]; 40.7%—Saitoh et al. [27]; 46%—Kiefer et al. [21]; 50.9%—Hosokawa et al. [16]; 61.2%—DesJardin et al. [28]; 89.5%—Blakeway et al. [13]). This was mostly due to the earlier stages of the COVID-19 pandemic while completing the data collection in the above-mentioned studies. Similarly, safety concerns and insufficient data were raised as one of the main reasons for rejection [5,8,27,29]. In the majority of available cohorts, resembling risk factors for vaccine rejection were mentioned—less formal education/lower socioeconomic status and multiparity are the most common ones [5,13,28]. This brings the conclusion that the reasons behind vaccine hesitancy are quite universal in different settings which may facilitate widespread campaigns for expecting mothers and those planning to get pregnant.

### 4.2. Strengths and Limitations

There are certain limitations to be mentioned. This study presents a sample of pregnant and postpartum women involved to a different extent in care at a single, centrally located unit, in a large urban center, with possible selection bias. Therefore, the study was offered mainly to an educated, modern, urban population, which might have been reflected in the vaccine acceptance rate and the conclusion could be drawn with limited generalizability. In terms of survey development, we did not use a validated questionnaire, but the formula was based rather on our previous study and clinical practice [17]. The acceptance was assessed at a single point in time, and it may be fluctuating throughout pregnancy—hesitating individuals could have eventually accepted the vaccination. As the Tdap vaccine is administered in the third trimester, some responders at the early stages of pregnancy might not have had a discussion on the matter with a healthcare professional. The questionnaire does not assess the rate of COVID-19 infection and complications among the closest family and friends, as well as personal beliefs and values. The study was totally dependent on patients’ self-reported vaccination status and the information was not verified in any databases.

Numerous assets prove the study as valuable research. In order to increase the number of responders, two methods of data collection were introduced. The online questionnaire was powered by the biggest platform for survey conduction in our country, which allowed the Internet version to be approachable and easily navigable on any electronic device for the responder’s convenience. The final number of interviewees exceeded a thousand, making it a considerable basis for analysis. The survey was fully anonymous, which helped in obtaining reliable answers. As the demographic structure shows, the responders were embedded in different backgrounds, including a whole range of ages, income levels, and pregnancy stages.

The study took place in the middle of the fourth wave of the pandemic and Poland is placed among the most COVID-stricken countries in Europe, with 15,478 confirmed cases per 100,000 to date [30]. Enrollment took place a few months after the release of numerous recommendations and all the participants had a chance to undergo vaccination.

## 5. Conclusions

Understanding factors facilitating the decision as well as the obstacles is critical for implementing prevention methods [31]. These factors may serve as the basis for further campaigns and interventions to increase COVID-19 vaccine acceptance, especially among vulnerable populations.

## Figures and Tables

**Figure 1 vaccines-10-01179-f001:**
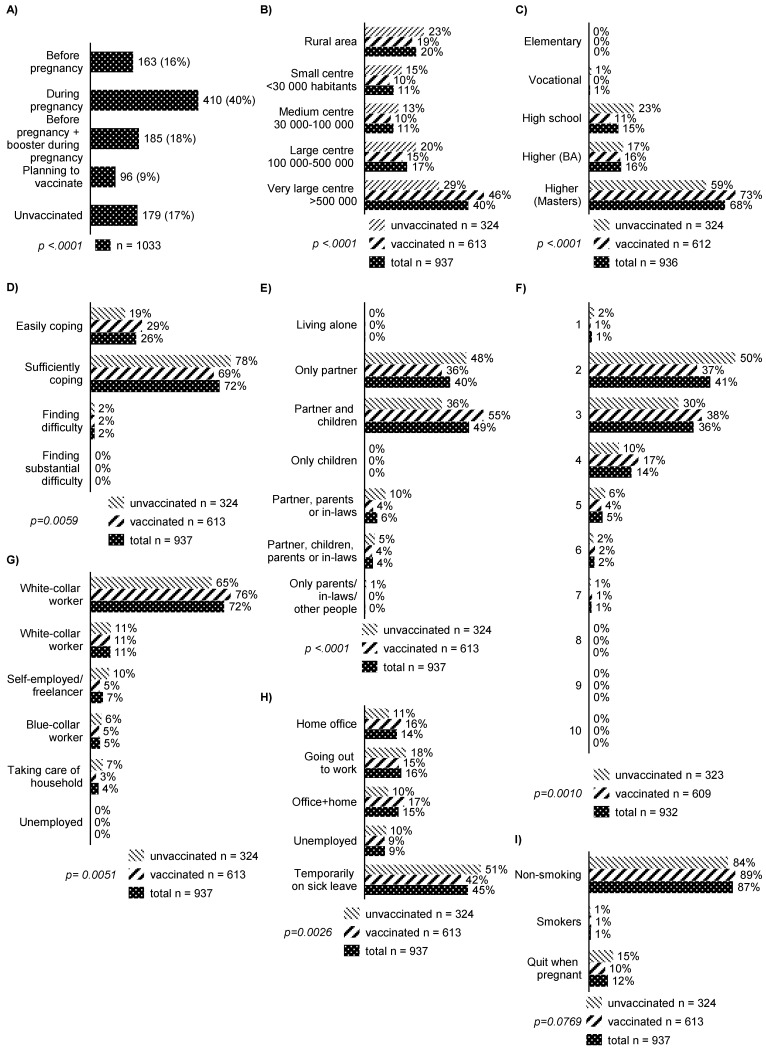
Questions asked in the survey. (**A**) Vaccination distribution among respondents; (**B**) Place of residence; (**C**) Education; (**D**) Financial situation; (**E**) Household description; (**F**) Number of people in the household; (**G**) Profession; (**H**) Current job status; (**I**) Smoking status; (**J**) Chronic conditions; (**K**) COVID-19 infection status; (**L**) Were you offered a COVID-19 vaccine by your doctor/midwife? (**M**) Vaccination first dose timing; (**N**) What was your doctor/midwife attitude towards COVID-19 vaccination? (**O**) Who helped you in making decision about vaccination? (**P**) Reason for vaccine rejection; (**Q**) Who did you experience negative attitude towards vaccination from? (**R**) Were you oferred a Tdap and/or influenza vaccine? (**S**) Did you doctor/midwife inform you about increased risk of COVID-19 infection in pregnancy?

**Table 1 vaccines-10-01179-t001:** Demographic characteristics of the study group.

	Total (N = 1033)	COVID-19 Vaccinated (N = 613)	COVID-19 Unvaccinated (N = 324)	Planning to Vaccinate to COVID-19 (N = 96)
%	100%	59.3%	31.4%	9.3%
Maternal age, mean (SD)	30.5 (4.1)	31.2 (3.9)	29.4 (4.2)	30.4 (4.1)
Mean gestational age (SD)	27.1 (9.1)	29.2 (8.4)	24.4 (9.1)	27.7 (11.1)
Mean gestational age at delivery (SD)	38.9 (2.4)	38.9 (2.6)	38.7 (2.2)	39.0 (1.4)
Number of pregnancies mean (SD)	1.68 (0.9)	1.7 (0.9)	1.63 (0.9)	1.71 (0.9)
Mean parity (SD)	0.89 (1.2)	0.94 (0.9)	1.73 (1.8)	1.09 (0.9)

**Table 2 vaccines-10-01179-t002:** Multivariate analysis of predictors of COVID-19 vaccine acceptance. OR: ordinal odds ratio. 95% CI: 95% confidence interval.

Effect	OR	95% CI	Standard Error	*p*-Value
Parity 0 vs. ≥1	1.74	1.43–2.12	0.0995	<0.0001
Place of residence (medium center vs. small + rural area)	1.15	0.7–1.88	0.2517	0.5852
Place of residence (large and very large center vs. small + rural area)	1.51	1.09–2.1	0.1673	0.0137
Education (masters + bachelor’s vs. elementary + vocational + high school)	2.26	1.53–3.34	0.1988	<0.0001
History of COVID-19 infection	1.92	1.43–2.59	0.1527	<0.0001
Vaccination offered by a doctor/midwife	1.54	1.11–2.12	0.1657	0.0097
Positive attitude of a doctor/midwife towards vaccination	2.74	1.2–6.29	0.4234	0.0172
Information from a doctor/midwife about a high risk of COVID-19 complications	2.15	1.55–2.98	0.1672	<0.0001

**Table 3 vaccines-10-01179-t003:** Multivariate analysis of predictors of COVID-19 vaccine rejection. OR: ordinal odds ratio. 95% CI: 95% confidence interval.

Effect	OR	95% CI	Standard Error	*p*-Value
Having children 3 and more vs. 0	4.8	2.0–11.6	0.4522	0.0006
Having children 3 vs. 1	2.15	0.89–5.2	0.4502	0.0896
Having children 3 vs. 2	1.78	0.69–4.59	0.4825	0.2318
Education (elementary + vocational + high school vs. master’s + bachelor’s)	2.29	1.55–3.39	0.1990	<0.0001
Lack of COVID-19 infection in the past	1.89	1.4–2.55	0.1533	<0.0001

## Data Availability

The data presented in this study are available on request from the corresponding author.

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
