# Peer review of "COVID-19 Vaccination Status among Pregnant and Postpartum Women—A Cross-Sectional Study on More Than 1000 Individuals"

_vaccines, 2022, doi:10.3390/vaccines10081179_

Round 1
Reviewer 1 Report
Authors should highlight the similarities and differences between similar studies and their work in the discussion section. In 2022, there are at least seven articles on this subject that the authors can cite.
Author Response
Dear Reviewer,
Thank you very much for all the comments, your feedback is much appreciated. We have added a new paragraph where results from other studies have been compared to our data (lines 276-286).
Kind regards,
Urszula Nowacka
Reviewer 2 Report
The manuscript (vaccines-1822573) entitled “COVID-19 vaccination status amongst pregnant and postpartum women – a cross-sectional study on more than 1000 individuals” by Dr. Nowacka (Department of Obstetrics and Gynecology, Institute of Mother and Child, Poland) reports on a prospective cross-sectional study conducted in a tertiary center for maternal - fetal medicine and obstetrics on the vaccine uptake and hesitancy rate, characteristics and factors contributing to a decision-making process among pregnant and postpartum individuals. Main result sindicate that women with higher level of education receiving positive feedback about vaccination or informed about COVID-19 complications in pregnancy were most likely to accept the vaccination. Hesitancy was associated with multiparity worse educational status and lack of previous COVID infection.
The manuscript will increase our knowledge behind the understanding factors behind COVID-19 vaccination status and its lack of acceptability/hesitancy in the general population, which is a very important topic. I believe that the work will therefore have an adequate impact in these fields.
Overall, the text is interesting, well written, clear and easy to follow. The main strength of the ms is the proposed experimental design which is well performed. Tables are fine and informative. I have made suggestions for improving figure 1
My final recommendation is therefore a minor revision. I have several, minor, observations/ comments for improving the quality of the manuscript:
Line 16 “COVID-19” sohuld be “Coronavirus disease 2019 (COVID-19)” when mentioned for the first itme. The same observation can be made for SARS-CoV-2 , line 38. Moreover, line 27 it should be “COVID-19”
Line 21 I discourage starting a sentence with a number
Lines38-42 Pregnant female are susceptible of viral infections, this susceptibility might negatively influence pregnancy outcome (PMID: 32854278 and PMID: 34961973). This important information and references should be included.
Lines 63-65 additional information on vaccine safety is here PMID: 34389291
Lines 106, 163 as well as other lines instead of individuals I would say participants and/or females
Supporting references should be included for the “statistical analysis” section
Figure 1 can be colored for a better reading. It should be helpful for the reader
4.1 and 4.2 subheads are unnecessary and can be removed for a better reading of the text
Line 236 “which is consistent?”
Author Response
Dear Reviewer,
Thank you very much for all the comments, your feedback is much appreciated. Please find all the answers below.
Kind regards,
Urszula Nowacka
- “Line 16 “COVID-19” sohuld be “Coronavirus disease 2019 (COVID-19)” when mentioned for the first itme. The same observation can be made for SARS-CoV-2 , line 38. Moreover, line 27 it should be “COVID-19”.”
- Changes have been made in the main text and abstract. We followed WHO recommendation (“coronavirus disease” = COVID-19). https://www.who.int/emergencies/diseases/novel-coronavirus-2019/technical-guidance/naming-the-coronavirus-disease-(covid-2019)-and-the-virus-that-causes-it#:~:text=Naming%20the%20coronavirus%20disease%20(COVID,the%20virus%20that%20causes%20it
- Line 21 I discourage starting a sentence with a number
- A change has been made.
- Lines38-42 Pregnant female are susceptible of viral infections, this susceptibility might negatively influence pregnancy outcome (PMID: 32854278 and PMID: 34961973). This important information and references should be included.
- We have mentioned this important issue (lines 38-40).
- Lines 63-65 additional information on vaccine safety is here PMID: 34389291
- We have included this study (line 64-65).
- Lines 106, 163 as well as other lines instead of individuals I would say participants and/or females
- Changes have been made.
- Supporting references should be included for the “statistical analysis” section.
- Supporting materials have been included (lines 157 and 161).
- Figure 1 can be colored for a better reading. It should be helpful for the reader
- Originally, this issue was of much consideration. Finally, as many people still print the manuscript on black and white printers, we wanted the graphs to remain readable and approachable in different settings.
- 1 and 4.2 subheads are unnecessary and can be removed for a better reading of the text
- Subheads have been removed.
- Line 236 “which is consistent?”
- A mistake has been removed.

Round 2
Reviewer 1 Report
I have no further comments